# Attention Bottlenecks for Multimodal Fusion

**Arsha Nagrani**   **Shan Yang**   **Anurag Arnab**   **Aren Jansen**
**Cordelia Schmid**   **Chen Sun**
{anagrani, shanyang, aarnab, arenjansen, cordelias, chensun}@google.com
Google Research

## Abstract

Humans perceive the world by concurrently processing and fusing high-dimensional inputs from multiple modalities such as vision and audio. Machine perception models, in stark contrast, are typically modality-specific and optimised for unimodal benchmarks, and hence late-stage fusion of final representations or predictions from each modality (*'late-fusion'*) is still a dominant paradigm for multimodal video classification. Instead, we introduce a novel transformer based architecture that uses 'fusion bottlenecks' for modality fusion at *multiple* layers. Compared to traditional pairwise self-attention, our model forces information between different modalities to pass through a small number of bottleneck latents, requiring the model to collate and condense relevant information in each modality and share what is necessary. We find that such a strategy improves fusion performance, at the same time reducing computational cost. We conduct thorough ablation studies, and achieve state-of-the-art results on multiple audio-visual classification benchmarks including Audioset, Epic-Kitchens and VGGSound. All code and models will be released.

## 1   Introduction

Simultaneous multimodal sensations are a crucial enabler of human perceptual learning [50]. For artificial learning systems, however, designing a unified model for modality fusion is challenging due to a number of factors: (i) variations in learning dynamics between modalities [56], (ii) different noise topologies, with some modality streams containing more information for the task at hand than others, as well as (iii) specialised input representations. The difference in input representations between audio and vision is particularly stark – many state of the art audio classification methods rely on short term Fourier analysis to produce log-mel spectrograms, often using them as inputs to CNN architectures designed for images [26, 48]. These time-frequency representations have different distributions to images – multiple acoustic objects can have energy at the same frequency, and the translation invariances of CNNs may no longer be a desired property (while an acoustic object can be shifted in time, a shift in frequency could alter the meaning entirely). In contrast, the visual stream in a video is three-dimensional (two spatial and one temporal), and while different spatial regions of the image correspond to different objects, there is the unique challenge of high redundancy across multiple frames. Hence input representations, and consequently neural network architectures and benchmarks tend to vary wildly for different modalities. For simplicity, the dominant paradigm for multimodal fusion therefore often consists of an ad-hoc scheme that involves integrating separate audio and visual networks via their output representations or scores i.e. 'late-fusion' [22, 44].

In this work, we present a new transformer based model for audiovisual fusion in video. Despite originally being proposed for NLP tasks, there has been recent interest in transformers [54] as universal perceptual models [29], due to their ability to model dense correlations between tokens, at the same time making few assumptions about their inputs (and because continuous perceptual inputs can be tokenised). By dividing dense continuous signals into patches and rasterising them

35th Conference on Neural Information Processing Systems (NeurIPS 2021).

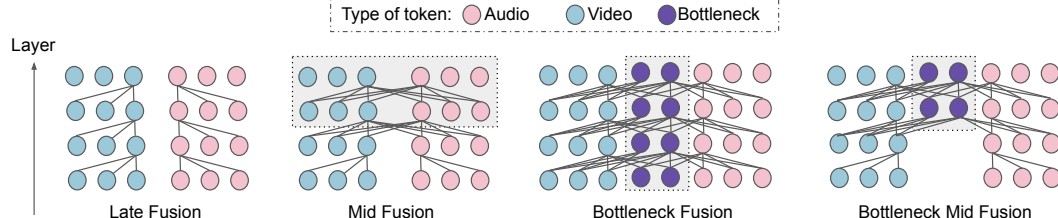

Figure 1: **Cross-modal Fusion**. Unlike late fusion (left), where no cross-modal information is exchanged in the model until after the classifier, we investigate two pathways for the exchange of cross-modal information. The first is via standard pairwise self attention across all hidden units in a layer, but applied only to later layers in the model – mid fusion (middle, left). We also propose the use of 'fusion bottlenecks' (middle, right) that restrict attention flow within a layer through tight latent units. Both forms of restriction can be applied in conjunction (Bottleneck Mid Fusion) for optimal performance (right). We show $B = 2$ bottleneck units and 3 hidden units per modality. Grey boxes indicate tokens that receive attention flow from both audio and video tokens.

to 1D tokens, transformers have been shown to perform competitively for image (ViT [16]) and video classification (ViViT [6]), and more recently, audio classification (AST [23]). Because these models are able to elegantly handle variable length sequences, a natural first extension would be to feed in a sequence of both visual and auditory patches to a transformer, with minimal changes to the architecture. This 'early fusion' model allows free attention flow between different spatial and temporal regions in the image, as well as across frequency and time in the audio spectrogram. While theoretically appealing, we hypothesise that full pairwise attention at all layers of the model is not necessary because audio and visual inputs contain dense, fine-grained information, much of which is redundant. This is particularly the case for *video*, as shown by the performance of 'factorised' versions of [6]. Such a model would also not scale well to longer videos due to the quadratic complexity of pairwise attention with token sequence length. To mitigate this, we propose two methods to restrict the flow of attention in our model. The first follows from a common paradigm in multimodal learning, which is to restrict cross-modal flow to later layers of the network, allowing early layers to specialise in learning and extracting unimodal patterns. Henceforth this is is referred to as 'mid fusion' (Fig. 1, middle left), where the layer at which cross-modal interactions are introduced is called the 'fusion layer'. The two extreme versions of this are 'early fusion' (all layers are cross-modal) and 'late fusion' (all are unimodal) which we compare to as a baselines. Our second idea (and main contribution), is to restrict cross-modal attention flow between tokens *within* a layer. We do this by allowing free attention flow within a modality, but force our model to collate and 'condense' information from each modality before sharing it with the other. The core idea is to introduce a small set of latent *fusion* units that form an 'attention bottleneck', through which cross-modal interactions within a layer must pass. We demonstrate that this 'bottlenecked' version, which we name Multimodal Bottleneck Transformer (*MBT*), outperforms or matches its unrestricted counterpart, but with lower computational cost.

Concretely, we make the following contributions: (i) We propose a new architecture (*MBT*) for audiovisual fusion. Our model restricts the flow of cross-modal information between latent units through tight fusion 'bottlenecks', that force the model to collect and 'condense' the most relevant inputs in each modality (and therefore share only that which is necessary with the other modality). This avoids the quadratic scaling cost of full pairwise attention, and leads to performance gains with less compute; (ii) We apply MBT to image and spectogram patches (Fig. 2), and explore a number of ablations related to the fusion layer, the sampling of inputs and data size; and finally (iii) We set the new state-of-the-art for video classification across a number of popular audio-visual benchmarks, including AudioSet [21], Epic-Kitchens100 [12] and VGGSound [10]. On the Audioset dataset, we outperform the current state of the art by 5.9 mAP (12.7% relative improvement).

## 2  Related work

**Audiovisual learning:** Audiovisual multimodal learning has a rich history, both before and during the deep learning era [47]. Given the limited available data and computational resources, early work focused on relatively simple early-stage (e.g. stacking hand-designed features) and late-stage

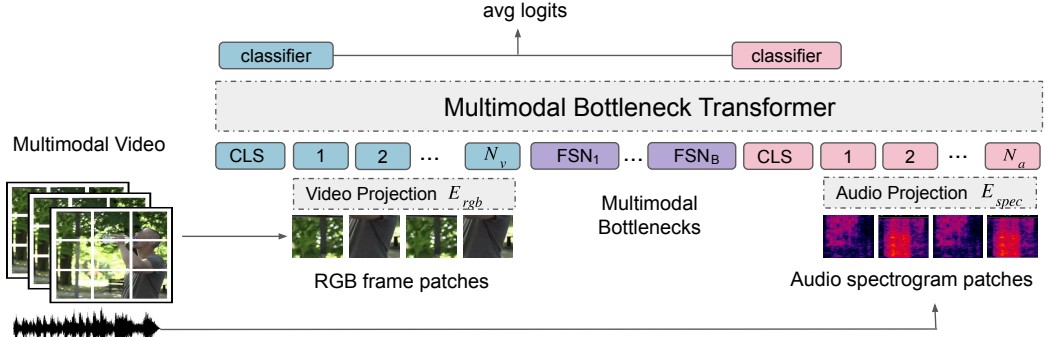

Figure 2: **A Multimodal Fusion Transformer applied to audiovisual inputs.** The input sequence consists of image and spectrogram patches. These are then projected into tokens and appended to special CLS (classification) and FSN (fusion bottleneck) tokens. Our transformer encoder then uses self attention to model unimodal information, and restricts cross-modal information flow via cross attention with the bottleneck tokens at multiple layers of the network.

(e.g. score fusion) techniques [11]. Deep learning has allowed more sophisticated strategies in which modality-specific or joint latents are implicitly learned to mediate the fusion. The result has enabled major advances in a range of downstream supervised audiovisual tasks [43, 34, 17]. In the supervised setting, multiple modality-specific convolution networks can be jointly trained, whose intermediate activations are then combined by summation [32] or via 'lateral connections' [57]. In the unsupervised setting, audiovisual learning is commonly used to learn good unimodal representations, with a popular pretraining task being to synchronise signals from different modalities via a contrastive loss [4, 5, 7, 44, 30, 2, 3], however each modality is usually encoded separately under this setup.

**Multimodal transformers:** The self attention operation of transformers provides a natural mechanism to connect multimodal signals. Multimodal transformers have been applied to various tasks including audio enhancement [17, 53], speech recognition [24], image segmentation [58, 53], cross-modal sequence generation [39, 37, 49], image and video retrieval [25, 20, 8], visual navigation [46] and image/video captioning/classification [41, 52, 51, 36, 28]. For many works, the inputs to transformers are the output representations of single modality CNNs [35, 20] – unlike these works we use transformer blocks throughout, using only a single convolutional layer to rasterise 2D patches. The tokens from different modalities are usually combined directly as inputs to the transformers [38], for example, the recently released Perceiver model [29] introduces an iterative attention mechanism which takes concatenated raw multimodal signals as inputs, which corresponds to our 'early fusion' baseline. In contrast, we carefully examine the impact of different modality fusion strategies, including limiting cross-modal attention flow to later layers of our model, and 'channeling' cross-modal connections through bottlenecks in our proposed Multimodal Bottleneck Transformer (MBT).

## 3 Multimodal fusion transformers

In this section we describe our proposed Multimodal Bottleneck Transformer (MBT). We begin by summarising the recently proposed Vision Transformer (ViT) [16] and Audio Spectrogram Transformer (AST) [23], developed for image and audio classification respectively, in Sec. 3.1. We then describe our extension to the audio-visual fusion case. We discuss three different token fusion strategies (Sec. 3.2), and finally discuss the fusion pathway in the entire model (Sec. 3.3), which involves restricting multimodal fusion to certain layers of the model.

### 3.1 The ViT and AST architectures

Vision Transformer (ViT) [16] (and a recent extension to audio – Audio Spectrogram Transformer (AST) [23]) adapts the Transformer architecture [54], originally designed for natural language processing, to process 2D inputs with minimal changes. The key insight is to extract $N$ non-overlapping patches from the RGB image (or the audio spectrogram), $x_i \in \mathbb{R}^{h \times w}$, and convert them into a series of 1D tokens $z_i \in \mathbb{R}^d$, as follows:

$$\mathbf{z} = g(\mathbf{x}; \mathbf{E}, z_{\text{cls}}) = [z_{\text{cls}}, \mathbf{E}x_1, \mathbf{E}x_2, ..., \mathbf{E}x_N] + \mathbf{p}. \tag{1}$$

Here, $\mathbf{E}$ is a linear projection mapping each token to $\mathbb{R}^d$, $z_{\text{cls}}$ is a special token prepended to this sequence so that its representation at the final layer can be passed to a classifier for classification tasks [15], and $\mathbf{p} \in \mathbb{R}^{(N+1) \times d}$ is a learned positional embedding added to the tokens to retain positional information (as all subsequent self-attention operations are permutation invariant).

The tokens are then passed through an encoder consisting of a sequence of $L$ transformer layers. Each transformer layer consists of Multi-Headed Self-Attention (MSA), Layer Normalisation (LN) and Multilayer Perceptron (MLP) blocks applied using residual connections. We denote a transformer layer, $\mathbf{z}^{l+1} = \text{Transformer}(\mathbf{z}^l)$ as

$$\mathbf{y}^l = \text{MSA}(\text{LN}(\mathbf{z}^l)) + \mathbf{z}^l \tag{2}$$

$$\mathbf{z}^{l+1} = \text{MLP}(\text{LN}(\mathbf{y}^l)) + \mathbf{y}^l. \tag{3}$$

Here, the MSA operation [54] computes dot-product attention [54] where the queries, keys and values are all linear projections of the same tensor, $\text{MSA}(\mathbf{X}) = \text{Attention}(\mathbf{W}^Q \mathbf{X}, \mathbf{W}^K \mathbf{X}, \mathbf{W}^V \mathbf{X})$. We further define Multi-Headed Cross Attention (MCA) between two tensors, $\mathbf{X}$ and $\mathbf{Y}$, where $\mathbf{X}$ forms the query and $\mathbf{Y}$ forms the keys and values which are used to reweight the query as $\text{MCA}(\mathbf{X}, \mathbf{Y}) = \text{Attention}(\mathbf{W}^Q \mathbf{X}, \mathbf{W}^K \mathbf{Y}, \mathbf{W}^V \mathbf{Y})$. This will be used in our multimodal case, as described next.

### 3.2 Multimodal transformer

We now describe our extension to the multimodal case. We begin by discussing three different token fusion strategies.

#### 3.2.1 Fusion via vanilla self-attention

We begin by describing a 'vanilla' fusion model, which simply consists of the regular transformer applied to multimodal inputs. Our method of tokenising video is straightforward – given a video clip of length $t$ seconds, we uniformly sample $F$ RGB frames and convert the audio waveform into a single spectrogram. We then embed each frame and the spectrogram independently following the encoding proposed in ViT [16], and concatenate all tokens together into a single sequence.

Formally, if we have extracted a total of $N_v$ RGB patches from all $F$ sampled frames, $\mathbf{x}_{\text{rgb}} \in \mathbb{R}^{N_v \times d}$, and $N_a$ spectrogram patches, $\mathbf{x}_{\text{spec}} \in \mathbb{R}^{N_a \times d}$, our sequence of tokens is

$$\mathbf{z} = [\mathbf{z}_{\text{rgb}} || \mathbf{z}_{\text{spec}}] \quad \text{where} \quad \mathbf{z}_{\text{rgb}} = g(\mathbf{x}_{\text{rgb}}; \mathbf{E}_{\text{rgb}}, z_{\text{cls-rgb}}) \quad \text{and} \quad \mathbf{z}_{\text{spec}} = g(\mathbf{x}_{\text{spec}}; \mathbf{E}_{\text{spec}}, z_{\text{cls-spec}}). \tag{4}$$

Here, $[\mathbf{z}_{\text{rgb}} || \mathbf{z}_{\text{spec}}]$ denotes the concatenation of the tokens for each modality. We use different projections $\mathbf{E}_{\text{rgb}}$ and $\mathbf{E}_{\text{spec}}$ for RGB and spectrogram patches respectively, and prepend a separate classification token for each modality.

Our multimodal encoder then applies a series of transformer layers in the same manner as above. Attention is allowed to flow freely through the network, i.e. each RGB token can attend to all other RGB and spectrogram tokens as follows: $\mathbf{z}^{l+1} = \text{Transformer}(\mathbf{z}^l; \theta)$ with model parameters $\theta$. Here Transformer refers to a standard transformer layer with vanilla self-attention blocks.

#### 3.2.2 Fusion with modality-specific parameters

We can generalise this model by allowing each modality to have its own dedicated parameters $\theta_{\text{rgb}}$ and $\theta_{\text{spec}}$, but still exchange information via the attention mechanism. For this purpose, we define a Cross-Transformer layer:

$$\mathbf{z}_{\text{rgb}}^{l+1} = \text{Cross-Transformer}(\mathbf{z}_{\text{rgb}}^l, \mathbf{z}^l; \theta_{\text{rgb}}) \tag{5}$$

$$\mathbf{z}_{\text{spec}}^{l+1} = \text{Cross-Transformer}(\mathbf{z}_{\text{spec}}^l, \mathbf{z}^l; \theta_{\text{spec}}),$$

where the Cross-Transformer employs the generalised cross-attention operation that takes two sets of inputs $\mathbf{z}_1$ and $\mathbf{z}_2$ that are not necessarily overlapping. This layer follows the original transformer layer with the difference being that Eq. 2 becomes

$$\mathbf{y}^l = \text{MCA}(\text{LN}(\mathbf{z}_1^l), \text{LN}(\mathbf{z}_2^l)) + \mathbf{z}_1^l. \tag{6}$$

Finally, note that we have explicitly defined the parameters, $\theta_{\text{rgb}}$ and $\theta_{\text{spec}}$ of the cross-transformer layers in Eq. 5 as they are different for each modality. However, when $\theta_{\text{rgb}}$ and $\theta_{\text{spec}}$ are equal, ($\theta_{\text{rgb}} = \theta_{\text{spec}} = \theta$), the computation defined in Eq. 5 is equivalent to Sec. 3.2.1.

### 3.2.3 Fusion via attention bottlenecks

In order to tame the quadratic complexity of pairwise attention, we next introduce a small set of $B$ fusion bottleneck tokens $\mathbf{z}_{\text{fsn}} = [z_{\text{fsn}}^1, z_{\text{fsn}}^2, \ldots, z_{\text{fsn}}^B]$ to our input sequence (see Fig. 2). The input sequence is now

$$\mathbf{z} = [\mathbf{z}_{\text{rgb}} || \mathbf{z}_{\text{fsn}} || \mathbf{z}_{\text{spec}}]. \tag{7}$$

We then restrict all cross-modal attention flow in our model to be via these bottleneck tokens. More formally for layer $l$, we compute token representations as follows:

$$[\mathbf{z}_i^{l+1} || \hat{\mathbf{z}}_{\text{fsn}_i}^{l+1}] = \text{Transformer}([\mathbf{z}_i^l || \mathbf{z}_{\text{fsn}}^l]; \theta_i) \tag{8}$$

$$\mathbf{z}_{\text{fsn}}^{l+1} = \text{Avg}_i(\hat{\mathbf{z}}_{\text{fsn}_i}^{l+1}) \tag{9}$$

Here $i$ indexes each modality, in this case RGB and Spec, and $\mathbf{z}_{\text{rgb}}$ and $\mathbf{z}_{\text{spec}}$ can only exchange information via the bottleneck $\mathbf{z}_{\text{fsn}}$ within a transformer layer. We first create modality specific temporary bottleneck fusion tokens $\hat{\mathbf{z}}_{\text{fsn}_i}$, which are updated separately and simultaneously with audio and visual information (Equation 8). The final fusion tokens from each cross-modal update are then averaged in Equation 9. We also experiment with asymmetric updates for the bottleneck tokens (see appendix) and found performance was robust to this choice. We keep the number of bottleneck tokens in the network to be much smaller than the total number of latent units per modality ($B \ll N_v$ and $B \ll N_a$). Because all cross-modal attention flow must pass through these units, these tight 'fusion' bottlenecks force the model to condense information from each modality and share that which is necessary. As we show in the experiments, this increases or maintains performance for multimodal fusion, at the same time reducing computational complexity. We also note that our formulation is generic to the type and the number of modalities.

## 3.3 Where to fuse: early, mid and late

The above strategies discuss fusion within a layer, and in most transformer architectures (such as ViT), every layer consists of an identical set of operations. A common paradigm in multimodal learning, however, is to restrict early layers of a network to focus on unimodal processing, and only introduce cross-modal connections at later layers. This is conceptually intuitive if we believe lower layers are involved in processing low level features, while higher layers are focused on learning semantic concepts – low-level visual features such as edges and corners in images might not have a particular sound signature, and therefore might not benefit from early fusion with audio [57].

This can be implemented with our model as follows: We initially perform vanilla self-attention among tokens from a single modality for $L_f$ layers. Thereafter, we concatenate all latent tokens together, $\mathbf{z}^{L_f} = [\mathbf{z}_{\text{rgb}}^{L_f} || \mathbf{z}_{\text{spec}}^{L_f}]$ and pass them through the remaining $L - L_f$ layers where the tokens are fused according to Sec. 3.2. Here, $L_f = 0$ corresponds to an 'early-fusion' model, $L_f = L$ a 'late-fusion' model, and $0 < L_f < L$ a 'mid-fusion' one. More formally, this can be denoted as

$$\mathbf{z}_{\text{rgb}}^{l+1} = \text{Transformer}(\mathbf{z}_{\text{rgb}}^l; \theta_{\text{rgb}}), \mathbf{z}_{\text{spec}}^{l+1} = \text{Transformer}(\mathbf{z}_{\text{spec}}^l; \theta_{\text{spec}}) \qquad \text{if } l < L_f$$

$$\mathbf{z}^l = [\mathbf{z}_{\text{rgb}}^l || \mathbf{z}_{\text{spec}}^l], \qquad\qquad \mathbf{z}^{l+1} = \text{Multimodal-Transformer}(\mathbf{z}^l; \theta_{\text{spec}}, \theta_{\text{rgb}}) \quad \text{otherwise}$$

where $\text{Multimodal-Transformer}(\cdot)$ can refer to either of the 3 fusion strategies described in Sec 3.2.

## 3.4 Classification

For all model variants described above, we pass output representations of the CLS tokens $z_{\text{cls-rgb}}^L$ and $z_{\text{cls-spec}}^L$ to the same linear classifier and average the pre-softmax logits.

# 4 Experiments

We apply MBT to the task of video classification. In this section we first describe the datasets used to train and test multimodal fusion and their respective evaluation protocols (Sec. 4.1), then discuss implementation details (Sec. 4.2). We then ablate the key design choices in our model (Sec. 4.3), before finally comparing our model to the state of the art (Sec. 4.4).

### 4.1 Datasets and evaluation protocol

We experiment with three video classification datasets – AudioSet [21], Epic-Kitchens-100 [12] and VGGSound [10], described in more detail below. Results on two additional datasets Moments in Time [42] and Kinetics [31] are provided in the appendix.

**AudioSet [21]** consists of almost 2 million 10-second video clips from YouTube, annotated with 527 classes. Like other YouTube datasets, this is a dynamic dataset (we only use the clips still available online). This gives us 20,361 clips for the balanced train set (henceforth referred to as mini-AudioSet or miniAS) and 18,589 clips for the test set. This test set is exactly the same as recent works we compare to, including Perceiver [29]. Instead of using the 2M unbalanced training set, we train on a (slightly more) balanced subset consisting of 500K samples (AS-500K). Details are provided in the appendix. Because each sample has multiple labels, we train with a binary cross-entropy (BCE) loss and report mean average precision (mAP) over all classes, following standard practice.

**Epic-Kitchens 100 [12]** consists of egocentric videos capturing daily kitchen activities. The dataset consists of 90,000 variable length clips spanning 100 hours. We report results for action recognition following standard protocol [12] - each action label is a combination of a verb and noun, and we predict both using a single network with two 'heads', both trained with a cross-entropy loss. The top scoring verb and action pair predicted by the network are used, and Top-1 action accuracy is the primary metric. Actions are mainly short-term (average length is 2.6s with minimum length 0.25s).

**VGGSound [10]** contains almost 200K video clips of length 10s, annotated with 309 sound classes consisting of human actions, sound-emitting objects and human-object interactions. Unlike AudioSet, the sound source for each clip is 'visually present' in the video. This was ensured during dataset creation through the use of image classifiers. After filtering clips that are no longer available on YouTube, we end up with 172,427 training and 14,448 test clips. We train with a standard cross-entropy loss for classification and report Top-1 and Top-5 classification accuracy.

### 4.2 Implementation details

Our backbone architecture follows that of ViT [16] identically, specifically we use ViT-Base (ViT-B, $L = 12$, $N_H = 12$, $d = 3072$)[1] initialised from ImageNet-21K [14], however we note that our method is agnostic to transformer backbone. Unless otherwise specialised, we use $B = 4$ bottleneck tokens for all experiments with bottleneck fusion. Bottleneck tokens are initialized using a Gaussian with mean of 0 and standard deviation of 0.02, similar to the positional embeddings in the public ViT [16] code. We randomly sample clips of $t$ seconds for training. RGB frames for all datasets are extracted at 25 fps. For AudioSet and VGGSound we sample 8 RGB frames over the sampling window of length $t$ with a uniform stride of length $(t \times 25)/8$. We extract $16 \times 16$ patches from each frame of size $224 \times 224$, giving us a total of $8 \times 14 \times 14 = 1568$ patches per video. For Epic-Kitchens (because the segments are shorter), we sample 32 frames with stride 1. Audio for all datasets is sampled at 16kHz and converted to mono channel. Similar to [23], we extract log mel spectrograms with a frequency dimension of 128 computed using a 25ms Hamming window with hop length 10ms. This gives us an input of size $128 \times 100t$ for $t$ seconds of audio. Spectrogram patches are extracted with size $16 \times 16$, giving us $50 \times 8 = 400$ patches for 8 seconds of audio. For images we apply the standard data augmentations used in [6] (random crop, flip, colour jitter), and for spectrograms we use SpecAugment [45] with a max time mask length of 192 frames and max frequency mask length of 48 bins following AST [23]. We set the base learning rate to 0.5 and train for 50 epochs, using Mixup [59] with $\alpha = 0.3$ and stochastic depth regularisation [27] with probability $p = 0.3$. All models (across datasets) are trained with a batch size of 64, synchronous SGD with momentum of 0.9, and a cosine learning rate schedule with warmup of 2.5 epochs on TPU accelerators using the Scenic library [13].

**Inference:** Following standard practice, we uniformly sample multiple temporal crops from the clip and average per-view logits to obtain the final result. The number of test crops is set to 4.

### 4.3 Ablation analysis

In this section we investigate the impact of the different architectural choices in MBT. Unless otherwise specified, we use the mini-AudioSet split for training and report results on the AudioSet eval split. More ablations on backbone size and pretraining initalisation can be found in the appendix.

---

[1] $L$ is the number of transformer layers, $N_H$ is the number of self-attention heads with hidden dimension $d$.

### 4.3.1 Fusion strategies

We implement all the three fusion strategies described in Sec. 3.2:
(i) **Vanilla self-attention** – Unrestricted pairwise attention between all latent units within a layer;
(ii) **Vanilla cross-attention with separate weights:** Same as above, but we now have separate weights for each modality. The latent units are updated via pairwise attention with all other latent units from both modalities; and finally
(iii) **Bottleneck fusion:** Here all cross-modal attention must pass through bottleneck fusion latents. Note that these three fusion strategies only describe attention flow between tokens within a layer. For strategies (ii) and (iii), we also conduct experiments showing the impact of restricting cross-modal attention to layers after a fixed fusion layer $L_f$. We investigate models with different fusion layers, $L_f = 0, 2, 4, 6, 8, 10, 12$, and present the results in Fig. 3.[2]
**Sharing weights for both modalities:** We first investigate the impact of sharing the encoder weights for both modalities (strategy (i) vs (ii)). The results can be found in Fig. 1 in the appendix. When modalities are fused at earlier layers, using separate encoders improves performance. For models with later fusion layers, performance is similar for both models. We hence use separate modality weights for further experiments.
**Fusion layer:** We then investigate the impact of varying the fusion layer $L_f$, for the latter two strategies: (ii) Vanilla Cross-Attention and (iii) Bottleneck Fusion. We conduct experiments with $L_f = 0, 2, 4, 6, 8, 10, 12$. We fix the input span $t$ to 4s and the number of bottleneck tokens $B$ to 4. We conduct 3 runs for each experiment and report mean and std deviation. As can be seen from Fig. 3 (left), 'mid fusion' outperforms both early ($L_f = 0$) and late fusion ($L_f = 12$), with optimal performance obtained by using fusion layer $L_f = 10$ for vanilla cross-attention and $L_f = 8$ for bottleneck attention. This suggests that the model benefits from restricting cross-modal connections to later layers, allowing earlier layers to specialise to learning unimodal features, however still benefits from multiple layers of cross-modal information flow. In appendix D , we confirm that mid fusion outperforms late fusion across a number of different datasets.
**Attention bottlenecks:** In Fig. 3, we also examine the effect of bottleneck attention vs vanilla cross-attention for multimodal fusion. We find that for all values of $L_f$ restricting flow to bottlenecks improves or maintains performance, with improvements more prominent at lower values of $L_f$. At $L_f = 10$, both perform similarly, note that at this stage we only have 3 fusion layers in the model. Our best performing model uses attention bottlenecks with $L_f = 8$, and we fix this for all further experiments. We also compare the amount of computation, measured in GFLOPs, for both fusion strategies (Fig. 3, right). Using a small number of bottleneck tokens (in our experiments $B = 4$) adds negligible extra computation over a late fusion model, with computation remaining largely constant with varying fusion layer $L_f$. This is in contrast to vanilla cross-fusion, which has a non-negligible computational cost for every layer it is applied to. We note that for early fusion ($L_f = 0$), bottleneck fusion outperforms vanilla cross-attention by over 2 mAP, with less than half the computational cost.
**Number of bottleneck tokens $B$:** We experiment with $B = 4, 36, 64, 256$ and $1024$, and find that performance is relatively consistent (all within 0.5 mAP). We hence fix the number of tokens to $B = 4$ for all experiments. It is interesting that with such a small number of cross-modal connections through only 4 hidden units ($B = 4$) at each cross-modal layer, we get large performance gains over late fusion (Fig. 3), highlighting the importance of allowing cross-modal information to flow at multiple layers of the model.

### 4.3.2 Input sampling and dataset size

In this section we investigate the impact of different modality sampling strategies. We also compare to single modality baselines – the visual-only and audio-only baselines consist of a vanilla transformer model applied to only the RGB or spectrogram patches respectively.
**Sampling window size $t$:** An advantage of our transformer based model is that we can easily input variable length token sequences. We experiment with varying the sampling window $t$ with the following values $t = 2, 4, 6$ and $8$ seconds (note that all videos in AudioSet are 10s), and show results in Fig. 4.[3] At inference, we uniformly sample multiple windows covering the entire video. While the number of spectrogram patches $N_a$ changes with $t$, we keep the number of RGB patches $N_v$

---

[2] Note that $L_f = 12$ refers to late fusion, where logits are only aggregated after the classifiers, and neither fusion strategy (ii) nor (iii) is applied, but we show results on the same plot for convenience.

[3] Averaged over 3 runs. Because error bars are small in the plot we also provide them in Table 3 in the appendix.

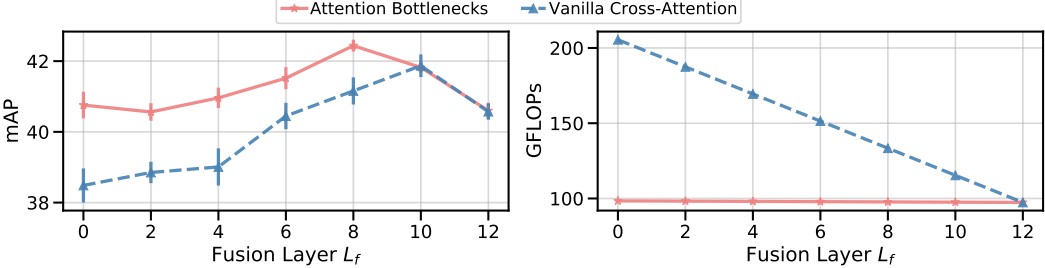

Figure 3: The impact of using attention bottlenecks for fusion on performance (left) and compute (right) at different fusion layers $L_f$ on AudioSet, using clip span $t = 4$ and $B = 4$ bottleneck tokens. Attention bottlenecks improve performance at lower computational cost.

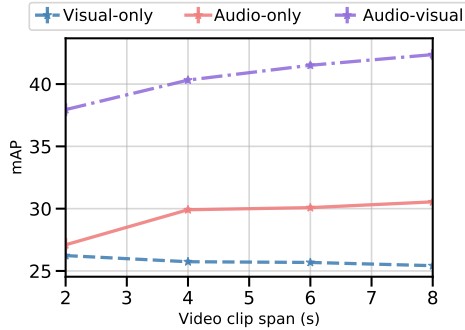

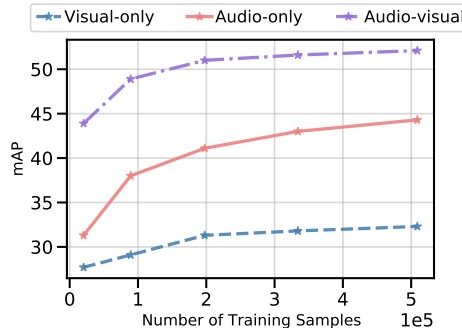

Figure 4: The effect of varying input clip span $t$ on the AudioSet test set.

Figure 5: The effect of training data size on the AudioSet test set.

fixed by changing the stride of frames (to avoid running out of memory). Our results indicate that the performance of both the audio and audio-visual fusion model increases with input span, however the performance of the visual-only model slightly decreases (we hypothesize that this is due to the increased fixed stride, meaning fewer frames are randomly sampled during training). We fix $t = 8s$ in all further experiments.

**Synchronous vs asynchronous sampling:** Given that auditory and visual events may not always be perfected aligned in videos [32], we also investigate asynchronous sampling of different modalities. Here input windows are sampled independently from the entire video clip for each modality. Results are provided in Fig. 2 in the appendix. We find performance to be largely robust to either case, and so for simplicity we use synchronised sampling for all further experiments.

**Modality MixUp:** While applying Mixup regularization [59] to training, we note that there are two different ways to apply it for multimodal inputs – the standard approach is to sample one set of mixup weights from a Beta distribution using the parameter $\alpha$, and use it to generate all virtual modality-label pairs [59]. We also explore a modified version which we call *modality mixup*, which samples an independent weight for each modality. Modality mixup imposes stronger augmentation than standard mixup, leading to a slight improvement (42.6 mAP to 43.9 mAP) on AudioSet.

**Impact of dataset size:** We show the impact of varying the number of training samples in Fig. 5, and find a monotonic increase with dataset size (more steeply for audio-only than visual-only).

### 4.4 Results

**Comparison to single modality performance:** We compare MBT to visual-only and audio-only baselines on AudioSet (Table 1), Epic-Kitchens (Table 2) and VGGSound (Table 3). Note we use the best parameters obtained via the ablations above, i.e. bottleneck fusion with $t = 8$, $B = 4$, $F_l = 8$ and modality mixup. For all datasets, multimodal fusion outperforms the higher-performing single modality baseline, demonstrating the value of complementary information. The relative importance of modalities for the classification labels varies (audio-only has higher relative performance for AudioSet and lower for Epic-Kitchens, while both audio and visual baselines are equally strong for VGGSound). This is (unsurprisingly) largely a function of the dataset annotation procedure and positions VGGSound as a uniquely suitable dataset for fusion. We also show that audio-visual fusion

| Model | Training Set | A only | V only | AV Fusion |
|-------|--------------|--------|--------|-----------|
| GBlend [56] | MiniAS | 29.1 | 22.1 | 37.8 |
| GBlend [56] | FullAS-2M | 32.4 | 18.8 | 41.8 |
| Attn Audio-Visual [18] | FullAS-2M | 38.4 | 25.7 | 46.2 |
| Perceiver [29] | FullAS-2M | 38.4 | 25.8 | 44.2 |
| MBT | MiniAS | 31.3 | 27.7 | 43.9 |
| MBT | AS-500K | **44.3** | **32.3** | **52.1** |

Table 1: **Comparison to SOTA on AudioSet [21].** We report mean average precision (mAP). We outperform works that train on the full Audioset (2M samples), while we train on only 500K samples.

| Model | Modalities | Verb | Noun | Action |
|-------|-----------|------|------|--------|
| Damen et al. [12] | A | 42.1 | 21.5 | 14.8 |
| AudioSlowFast [33]† | A | 46.5 | 22.78 | 15.4 |
| TSN [55] | V, F | 60.2 | 46.0 | 33.2 |
| TRN [60] | V, F | 65.9 | 45.4 | 35.3 |
| TBN [32] | A, V, F | 66.0 | 47.2 | 36.7 |
| TSM [40] | V, F | **67.9** | 49.0 | 38.3 |
| SlowFast [19] | V | 65.6 | 50.0 | 38.5 |
| MBT | A | 44.3 | 22.4 | 13.0 |
| MBT | V | 62.0 | 56.4 | 40.7 |
| MBT | A, V | 64.8 | **58.0** | **43.4** |

Table 2: **Comparison to SOTA on EpicKitchens-100 [12].** Modalities are **A:** Audio, **V:** Visual, **F:** Optical flow. †Uses pretraining on VGGSound.

provides slight performance gains for traditionally video only datasets such as Kinetics and Moments in Time (details provided in Appendix C ). We also examine per-class performance on the Audioset dataset (Figures 3 and 4 in the Appendix), and find that for the top 60 classes (ranked by overall performance), audio-visual fusion improves performance over audio only or visual only for almost all (57 out of 60) classes, except for 'bagpiping', 'emergency vehicle' and 'didgeridoo' which have strong audio signatures. For classes such as 'bicycle' and 'shuffling cards' where audio signals are weaker, fusion improves over the audio-only baseline by over 60% in absolute AP.

**Comparison to state of the art:** We compare MBT to previous fusion methods on AudioSet in Table 1. We outperform all previous works on fusion (even though we only train on a quarter of the training set – 500K samples), including the recently introduced Perceiver [29] which uses early fusion followed by multiple self attention layers, and Attn Audio-Visual [18] which uses self-attention fusion on top of individual modality CNNs. We compare to previous video classification methods on Epic-Kitchens in Table 2, and note that our model outperforms all previous works that use vision only, as well as TBN [32] which uses three modalities - RGB, audio and optical flow. Given VGGSound is

| Model | Modalities | Top-1 Acc | Top-5 Acc |
|-------|-----------|-----------|-----------|
| Chen et al‡ [10] | A | 48.8 | 76.5 |
| AudioSlowFast‡ [33] | A | 50.1 | 77.9 |
| MBT | A | 52.3 | 78.1 |
| MBT | V | 51.2 | 72.6 |
| MBT | A,V | **64.1** | **85.6** |

Table 3: **Comparison to the state of the art on VGGSound [10].** Modalities are **A:** Audio, **V:** Visual, **F:** Optical flow. ‡ We calculate metrics on our test set for a fair comparison using the scores provided by the authors.

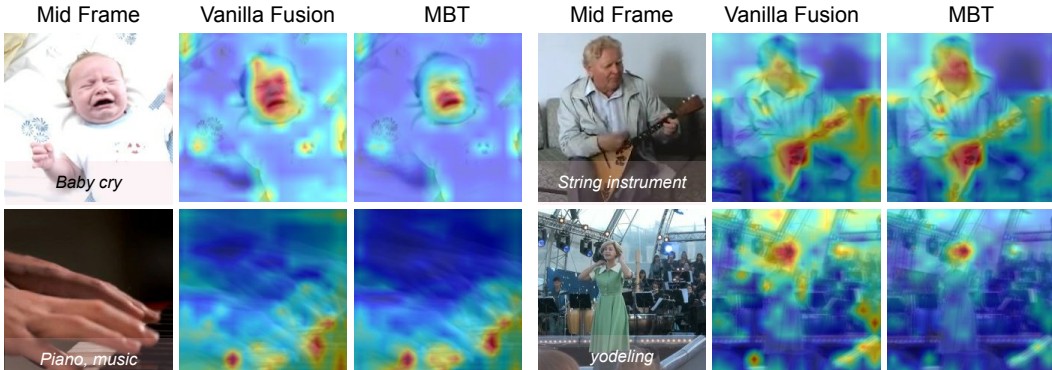

| Mid Frame | Vanilla Fusion | MBT | Mid Frame | Vanilla Fusion | MBT |

Figure 6: **Attention Maps.** We compute maps of the attention from the output CLS tokens to the RGB image input space for a vanilla self-attention model and MBT on the Audioset test set. For each video clip, we show the original middle frame on the left with the ground truth labels overlayed at the bottom. The attention is particularly focused on sound source regions in the video that contain motion, eg. the fingertips on the piano, the hands on the string instrument, faces of humans. The bottlenecks in MBT further force the attention to be localised to smaller regions of the images (i.e the mouth of the baby on the top left and the mouth of the woman singing on the bottom right).

a relatively new dataset, we compare to two existing audio-only works[4] (Table 3), and set the first audiovisual benchmark (that we are aware of) on this dataset.

**Visualisation of attention maps** Finally, we compute maps of the attention from the output CLS tokens to the RGB image input space using Attention Rollout [1]. Results on test images for both a vanilla fusion model and MBT trained on Audioset-mini (fusion layer $L_f = 8$) are shown in Figure 6. We show the attention maps summed over all the frames in the video clip. We note that first, the model focuses on semantically salient regions in the video for audio classification, particularly regions where there is motion that creates or modifies sound, i.e. the mouth of humans making sounds, fingertips on a piano, hands and instruments. This is unlike state of the art sound source localisation techniques trained with images [9], which tend to highlight the entire object. We further note that the attention maps for MBT are more localised to these regions, showing that the tight bottlenecks do force the model to focus only on the image patches that are actually relevant for the audio classification task and which benefit from early fusion with audio.

## 5 Conclusion

We propose a new transformer architecture (*MBT*) for audiovisual fusion, and explore a number of different fusion strategies using cross-attention between latent tokens. We propose a novel strategy to restrict cross-modal attention via a small set of fusion 'bottlenecks', and demonstrate that this improves performance over vanilla cross-attention at lower computational cost, achieving state of the art results on a number of benchmarks. Future work will involve extending MBT to other modalities such as text and optical flow.

**Limitations:** The fusion layer is a hyperparameter and may need to be tuned specifically for different tasks and datasets. We also only explore fully supervised fusion, and future work will tackle extensions to a self-supervised learning framework.

**Broader impact:** Multimodal fusion strategies are important for machine learning, as fusing complementary information from different modalities can increase robustness when applied to real world applications. We also note that transformers are in general compute-heavy, which can have adverse environmental effects. We propose a token fusion method via bottlenecks that helps reduce computational complexity when applying transformers for multimodal fusion. Finally, we observe that training datasets contain biases that may render models trained on them unsuitable for certain applications. It is thus possible that people use classification models (intentionally or not) to make decisions that impact different groups in society differently, and it is important to keep this in mind

---

[4]To fairly compare to these works, we obtain the scores on the full VGGSound test set from the authors, and compute accuracy metrics on our slightly smaller test set as described in Sec. 4.1.

when deploying, analysing and building upon these models.

**Acknowledgements:** We would like to thank Joao Carreira for helpful discussions on the Perceiver [29].

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
