# Attention Bottlenecks for Multimodal Fusion - Supplementary Materials

**Arsha Nagrani    Shan Yang    Anurag Arnab    Aren Jansen**
**Cordelia Schmid    Chen Sun**
{anagrani, shanyang, aarnab, arenjansen, cordelias, chensun}@google.com

Google Research

Here we provide additional ablation results on mini-Audioset (Sec. A) as well as analyse the per-class average precision of fusion over single modality baselines (Sec. B). We then provide results on two additional datasets, Moments in Time and Kinetics in Sec. C and perform some preliminary transfer learning experiments in Sec. E. Finally we provide details on the AS-500K split.

## A    Ablations on mini-Audioset

In this section we expand on the ablations provided in Sec. 4.3 of the main paper. Unless otherwise specified, ablations are performed using Audioset-mini as the training set and the Audioset test set for evaluation. For most experiments we conduct 3 runs and report mean and standard deviation.

### A.1    Symmetric vs asymmetric bottleneck updates

We also experiment with an asymmetric bottleneck update. This involves replacing Eq. 8 and 9 with the following:

$$[\mathbf{z}_{\mathrm{rgb}}^{l+1}||\hat{\mathbf{z}}_{\mathrm{fsn}}^{l+1}] = \mathrm{Transformer}([\mathbf{z}_{\mathrm{rgb}}^{l}||\mathbf{z}_{\mathrm{fsn}}^{l}]; \theta_{\mathrm{rgb}}) \tag{1}$$

$$[\mathbf{z}_{\mathrm{spec}}^{l+1}||\mathbf{z}_{\mathrm{fsn}}^{l+1}] = \mathrm{Transformer}([\mathbf{z}_{\mathrm{spec}}^{l}||\hat{\mathbf{z}}_{\mathrm{fsn}}^{l+1}]; \theta_{\mathrm{spec}}) \tag{2}$$

Here the bottleneck tokens are updated twice, first with visual information (Equation 1), and then with audio information (Equation 2). We also experimented with updating the bottlenecks with audio information first and compare both variations to the symmetric update in Table 1. We find performance is robust to all variations.

| RGB first | Spec first | Symmetric updates |
|:---:|:---:|:---:|
| 43.42±0.19 | 43.23±0.12 | 43.66±0.26 |

Table 1: Asymmetric vs symmetric bottleneck updates.

### A.2    Backbone architecture

We experiment with three standard ViT [5] backbones, ViT-Small, ViT-Base and ViT-Large on both Audioset-mini and VGGSound. We report results in Table 2 for audiovisual fusion with our best MBT model. We find that performance increases from ViT-Small to ViT-Base, but then drops for ViT-Large. This could be due to the fact that these datasets are on the smaller side, and more data might be required to take advantage of larger models.

35th Conference on Neural Information Processing Systems (NeurIPS 2021).

| Backbone | AS-mini | VGGSound |
|----------|---------|----------|
| ViT-Small | 38.2 | 59.0 |
| ViT-Base | 43.3 | 64.1 |
| ViT-Large | 42.2 | 61.4 |

Table 2: Performance with varying backbones on AS-mini and VGGSound.

### A.3 The impact of weight sharing

We investigate the impact of sharing the encoder weights for both modalities (strategy (i) vs (ii)) as described in Sec. 4.3.1 . Results are provided in Fig. 1 for different fusion layers $L_f$. When modalities are fused at earlier layers, using separate encoders improves performance. For models with later fusion layers, performance is similar for both models.

### A.4 Input sampling

Here we investigate asynchronous sampling of different modalities (where input windows are sampled independently from the entire video clip for each modality) as compared to synchronous sampling. Results are provided in Fig. 2 for different input span lengths $t$. Over multiple runs we find that performance is largely robust to either sampling choice. We hypothesise that asynchronous sampling provides the following trade-off: while it introduces a misalignment between the two modality inputs, slight shifts are also a good source of temporal augmentation. As the video clip span length grows, the possible options for misalignment between inputs are less severe, while the impact of additional augmentation is more evident.

In Table 3, we provide the results in numerical form used to create Fig. 4 . We perform 3 runs per experiment and report mean and standard deviation. All segments in AudioSet are 10 seconds long.

| Span Length $t$ | 2s | 4s | 6s | 8s |
|-----------------|-----|-----|-----|-----|
| Visual only | 26.23±0.16 | 25.74±0.18 | 25.68±0.02 | 25.43±0.02 |
| Audio only | 27.10±0.54 | 29.91±0.21 | 30.08±0.21 | 30.55±0.22 |
| Audio-Visual | 37.95±0.51 | 40.32±0.20 | 41.51±0.24 | 42.37±0.44 |

Table 3: The effect of varying input clip span $t$ on performance.

## B Per class performance

We also examine per-class average precision (AP) results for our best model trained on the mini-Audioset (note that this dataset has 527 classes). We first show the results for the 60 top ranked

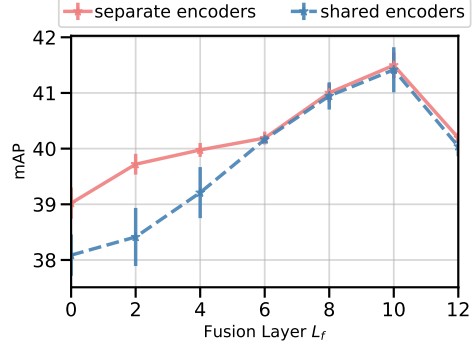

Figure 1: The effect of sharing weights for vanilla fusion.

Figure 2: Asynchronous vs synchronous sampling of RGB and spectrogram inputs.

classes in Audioset (by audio-visual mAP performance) in Fig. 3. We show the per class AP using our best fusion model (MBT), as well as the performance of audio only and visual only baselines. Audio-visual fusion improves performance over audio only or visual only for almost all (57 out of 60) classes, except for 'bagpiping', 'emergency vehicle' and 'didgeridoo' which have strong audio signatures. We then analyse the top 60 classes for which fusion has the largest improvement over single modality performance, over audio-only (Figure 4, top) and visual-only (Figure 4, bottom). For some classes such as 'bicycle' and 'shuffling cards', fusion improves over the audio-only baseline by over 60% in absolute AP. The class that benefits most from audio-visual fusion over a visual-only baseline is 'Whistling' (almost 80% improvement in absolute AP).

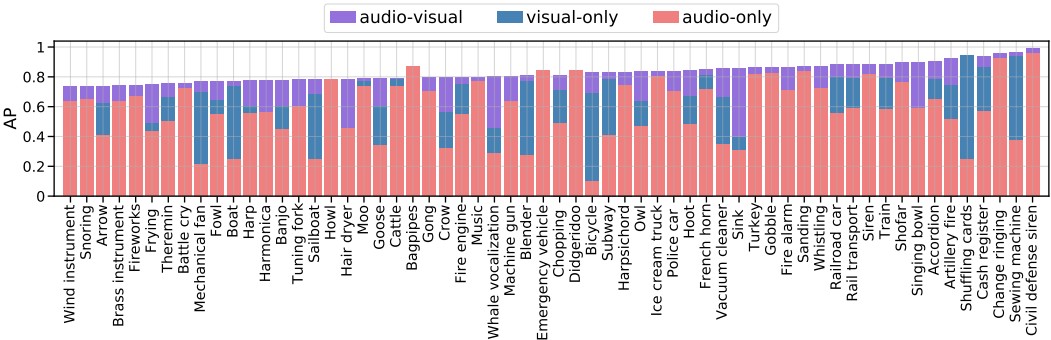

Figure 3: Per-class average precision for the top 60 classes in Audioset ranked by mAP. Best viewed in colour and zoomed in. Note how audio-visual fusion helps improve performance over audio only for almost all classes. The visual only model performs well for classes that have a stronger visual signature than audio, eg 'bicycle', 'mechanical fan', 'boat' and 'arrow'.

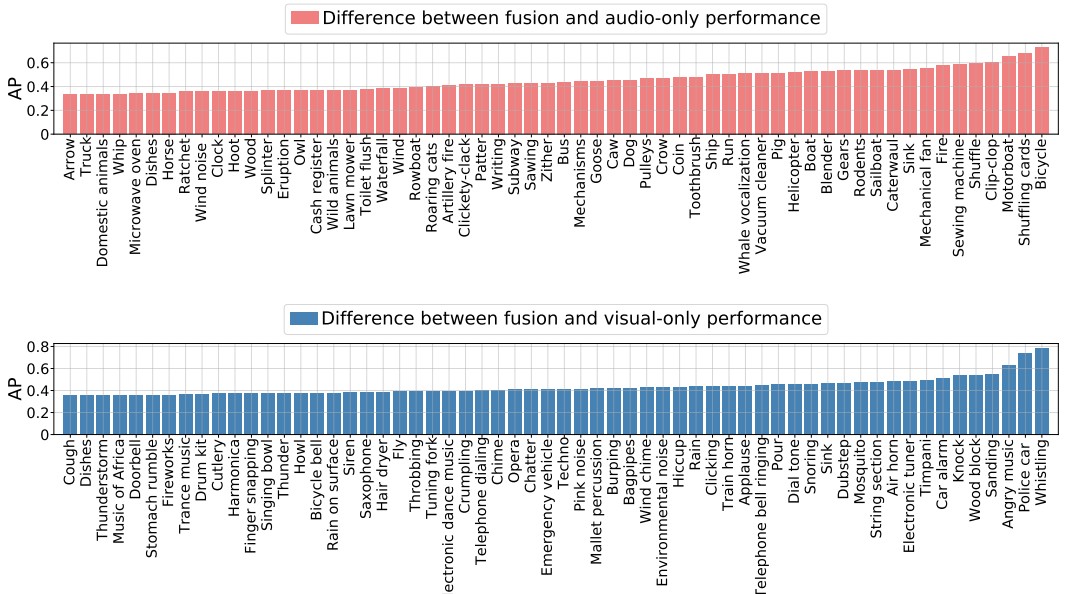

Figure 4: Top 60 classes that have the highest gain with fusion over a audio only (top) and visual only (bottom) baseline. Note how fusion improves the per class AP for certain classes by over 50% over a unimodal model. As expected, the classes that benefit most from visual information are 'bicycle' and 'shuffling cards' and the class that benefits most from audio is 'Whistling'.

## C  Additional Datasets

In this section we report results on 2 additional datasets, Moments in Time [11] and Kinetics [9].

### C.1  Moments In Time

Moments In Time [11] consists of 800,000, 3-second clips from YouTube videos. The videos are diverse and capture dynamic scenes involving animals, objects, people, or natural phenomena. The videos are labelled with 330 verb classes, each associated with over 1,000 videos. We show results for MBT compared to single modality baselines in Table 4. Our first observation is that audio-only performance is much lower than visual-only. This is largely a function of the annotation procedure for the dataset, however we also note that clips are only 3 seconds long, and as shown in Fig. 4 , audio-only performance is heavily dependant on the span length $t$ on Audioset, suggesting that it may be difficult to recognise audio events from shorter inputs. Our fusion model provides a further modest 1% boost to performance over the visual-only baseline.

| Model | Top-1 acc | Top-5 acc |
|---|---|---|
| I3D [4] | 29.5 | 56.1 |
| blVNet [6] | 31.4 | 59.3 |
| AssembleNet-101 [13] | 34.3 | 62.7 |
| ViViT-Base [2] | 37.3 | **64.2** |
| Ours (Audio-only) | 8.2 | 18.2 |
| Ours (Visual-only) | 36.3 | 59.3 |
| **MBT (AV)** | **37.3** | 61.2 |

Table 4: Comparison to state of the art on Moments in Time [11]. We report top 1 and top 5 classification accuracy. **AV:** Refers to audio-visual fusion.

### C.2  Kinetics

Kinetics [9] consists of 10-second videos sampled at 25fps from YouTube. We evaluate on both Kinetics 400 [9] and a commonly used subset Kinetics-Sound [1], containing 400 and 36 classes respectively. As these are dynamic datasets (videos may be removed from YouTube), we train and test on 209,552 and 17,069 videos respectively for Kinetics and report results on 1,165 videos for Kinetics-Sound. Results for MBT compared to single modality baselines are shown in Table 5. We note that on the entire Kinetics test set, our fusion model outperforms the visual only baseline by about 1% in top 1 accuracy (in line with other works [14] that demonstrate that audio for the large part does not improve performance for most Kinetics classes). This gap is widened, however, for the Kinetics-Sound subset of the dataset (over 4%), as expected because this subset consists of classes in Kinetics selected to have a strong audio signature [1].

## D  Dataset Variations for MBT vs Late Fusion

In this section we further analyse the significance of our method across all the popular video classification datasets used in the paper (most ablations results are only shown for mini-Audioset in the main paper). We note that the gap between MBT and late-fusion is highly dataset dependant (see Table 6), with our method providing an even greater advantage for Epic-Kitchens (almost 6% difference in Top 1 action accuracy).

## E  Transfer learning

We use checkpoints pretrained on VGGSound, Kinetics400 and AS-500K and finetune them on Audioset-mini and VGGSound (note we use a ViT-B backbone for these experiments, and report results for audiovisual fusion with our best MBT model). Results are provided in Table 7. While Kinetics400 pretraining gives a slight 0.7% mAP boost on AS-mini, VGGSound initialisation gives a

| Model | Kinetics | | Kinetics-Sounds | |
|---|---|---|---|---|
| | Top-1 | Top-5 | Top-1 | Top-5 |
| blVNet [6] | 73.5 | 91.2 | - | - |
| STM[8] | 73.7 | 91.6 | - | - |
| TEA [10] | 76.1 | 92.5 | - | - |
| TS S3D-G [15] | 77.2 | 93.0 | - | - |
| 3-stream SATT [3] | 77.7 | 93.2 | - | - |
| AVSlowFast, R101 [14] | 78.8 | 93.6 | 85.0† | - |
| LGD-3D R101 [12] | 79.4 | 94.4 | - | - |
| SlowFast R101-NL [7] | 79.8 | 93.9 | - | - |
| ViViT-Base [2] | 80.0 | 94.0 | - | - |
| Ours (Audio-only) | 25.0 | 43.9 | 52.6 | 71.5 |
| Ours (Visual-only) | 79.4 | 94.0 | 80.7 | 94.9 |
| **MBT (AV)** | **80.8** | **94.6** | **85.0** | **96.8** |

Table 5: Comparison to state of the art on Kinetics [9] and Kinetics Sound [1]. We report top-1 and top-5 classification accuracy. **AV:** Refers to audio-visual fusion. † Note the Kinetics-Sound test set has reduced since this work as videos have been removed from YouTube, hence this is not a direct comparison.

| Dataset | mini-Audioset | Epic-Kitchens | VGGSound | Moments in Time | Kinetics |
|---|---|---|---|---|---|
| Late Fusion | 41.80 | 37.90 | 63.3 | 36.48 | 77.0 |
| MBT | 43.92 | 43.40 | 64.1 | 37.26 | 80.8 |

Table 6: MBT vs late Fusion for different datasets. For each dataset we report the widely used primary metric, i.e. Audioset: mAP, Epic-Kitchens: Top-1 action accuracy, VGGSound, Moments in Time and Kinetics: Top-1 classification accuracy.

substantial 3% mAP boost over Imagenet Initialisation. On VGGSound, AS500K pretraining gives a more modest boost of 1.2% Top 1 Acc, while Kinetics pretraining does not help (expected as VGGSound is a larger dataset).

| Initialisation Checkpoint | AS-mini | VGGSound |
|---|---|---|
| ImageNet init. | 43.3 | 64.1 |
| VGGSound init. | 46.6 | N/A |
| K400 init. | 44.0 | 64.0 |
| AS-500K init. | N/A | 65.3 |

Table 7: Transfer learning on Audioset-mini and VGGSound.

## F    AS-500K details

The original unbalanced AudioSet training set consists of almost 2M samples, and is extremely unbalanced with most samples either labelled as speech or music. To improve training efficiency, we create a slightly more balanced subset called AudioSet-500K. The main issue is that AudioSet is multilabel, and this makes balancing difficult. We create AS-500K by greedily restricting the maximum number of samples per class to be 200K. Given the distribution of labels, this gives us a total size of 508,994 samples. We provide the full histogram of labels in Fig. 5 (note the number of samples is on a $\log_{10}$ scale).

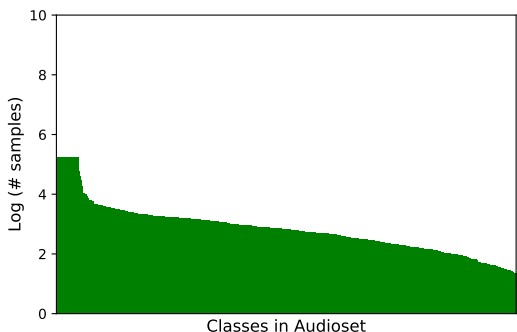

Figure 5: Class label histogram in the AudioSet-500K split.