# OpenReview forum: "Attention Bottlenecks for Multimodal Fusion"
_NeurIPS.cc/2021/Conference — NeurIPS 2021 Poster_

### Official Review · Reviewer_TS5J · 2021-07-01

**Rating:** 7
**Confidence:** 4

**Summary:**

For efficient multimodal fusion, the authors proposed the attention bottlenecks for audio-visual classification tasks. Compared to previous works, they emphasize their bottleneck architecture "to collate and condense the most relevant information in each modality." They showed that it also has the advantage of a lower computational cost. They evaluate the proposed method using three competitive methods, vanilla self-attention, vanilla cross-attention, and their attention bottlenecks. They noted that the position of modality fusion was a critical factor among early, mid and late.

**Limitations And Societal Impact:**

L1) I suggest updating the title, ".. for Multimodal Fusion" since this work limits to bi-modal and audio-visual classification tasks as the authors mentioned in the Conclusion. They fail to show how to scale to more than two modality tasks and if it is also effective other than audio-visual classification tasks.

L2) In the Abstract, the authors argued that the bottleneck forces "to collate and condense the most relevant information in each modality and only share what is necessary." Since the authors evaluate the method using macro metrics for the task, I suggest toning down to "most relevant" -> "relevant" and "only share" -> "share". The current explanation seems to be too strong without sufficient evidence.

L3) Although the attention bottlenecks have the advantage of computational cost, this is sensitive to the fusion layer $L_f$. In some cases, the proposed method, attention bottlenecks are indifferent from vanilla cross-attention as shown in Figure 3. This means if we adjust $L$ for the smaller or larger dataset, the hyperparameter $L_f$ would be critical to its performance. The authors omit this aspect in the Limitations of the Conclusion section.

**Main Review:**

- They showed that they can "keep the number of bottleneck tokens in the network to be much smaller than the total number of latent units per modality." I believe this is a novel observation and very interesting.
- I appreciate the explicit paragraph for the limitations.


### Clarity

c1) L111, could you more elaborate on the functionality of the special token $z_{cls}$ for readers why it is especially needed?

c2) L122, "MCA" is appeared without definition.

c3) L130, which interval of the audio waveform is converted to a single spectrogram? Could you clarify specifically?

c4) L193, what do you mean by a "slightly more" balanced subset used? Slightly more than 500k samples? Could you provide the details?

c5) L222, how do you get "80 x 5 = 400" patches for 8 seconds?

c6) Footnote 2, what do you mean by "... and neither strategy (ii) nor (iii) is applied"? (what is the subject and to what?)

c7) L296, for the modality mixup, how do you choose the weight for the labels, as in the Mixup? The explanation for the modality mixup seems to be insufficient to understand correctly.


### Minors

m1) "All (section) headings should be lower case except for the first word and proper nouns" according to the official style guideline.

m2) L56, "a baselines" -> "baselines"

m3) L217, "8 x 14 x 14" -> "8 x 16 x 16", since you extract 16 x 16 patches from each frame?

----
I appreciate your thorough feedbacks. After reading other reviews, I cannot resist to increase my score to 7.

**Time Spent Reviewing:**

6

---

> ### Author Response · Authors · 2021-08-10
> **Response to Reviewer TS5J**
>
> Thank you for your constructive and positive feedback
>
> **c1) L111, could you more elaborate on the functionality of the special token for readers why it is especially needed?**
>  The classification token is used to summarise all the tokens for a modality to be passed to the classification head, as done in the original BERT paper [11].
>
> **c2) L122, "MCA" is appeared without definition.**
>  MCA refers to MultiHead Cross Attention, and the formulaic definition is presented in L122 itself. We will clarify the expansion of the abbreviation in the paper.
>
> **c3) L130, which interval of the audio waveform is converted to a single spectrogram? Could you clarify specifically?**
>  From the input video, we randomly sample a video clip of $t$ seconds. The audio for this $t$ second clip is then used to extract a single spectrogram. Unless otherwise specified, we set $t=8$ seconds. We will clarify this in the paper.
>
> **c4) L193, what do you mean by a "slightly more" balanced subset used? Slightly more than 500k samples? Could you provide the details?**
>  This subset was created greedily by restricting the maximum number of samples per class to be 200K. The main issue is that AudioSet is multilabel, and this makes balancing difficult. Given the distribution of labels, this gives us a total size of 508,994 samples. Because this subset is also heavily unbalanced, we say that it is only “slightly” more balanced than the original AudioSet training set. We provide the full histogram of labels in Fig. 3 (note the number of samples is on a $\text{log}_{10}$ scale) at the anonymous link: https://11137mbt.github.io/11137/. We will add this description to Sec. 4.1 and the histogram to the appendix of the paper.
>
> **c5) L222, how do you get "80 x 5 = 400" patches for 8 seconds?**
>  The size of our spectrograms is 100 x 128 for 1 second of audio, and therefore 800 x 128 for 8 seconds. Because our patch size is 16, this gives (800/16) x (128/16) = 50 x 8 = 400 patches. We will rewrite it like this to be explicit in the paper, apologies that the way we had written it earlier was misleading.
>
> **c6) Footnote 2, what do you mean by "... and neither strategy (ii) nor (iii) is applied"? (what is the subject and to what?)**
> Footnote 2 refers to the late fusion numbers in Fig. 3 (fusion layer = 12). In this case, neither vanilla fusion nor bottleneck fusion is applied. There is no multimodal fusion in the encoder at all, and we simply average the pre-softmax logits for each modality. We will update the description to clarify this better in the paper.
>
> **c7) L296, for the modality mixup, how do you choose the weight for the labels, as in the Mixup?**
>  The weights are sampled from a Beta distribution (as is done in standard Mixup regularization [51]). The difference with standard Mixup is that modality Mixup samples a separate weight for each modality.
>
> **m1 and m2)** Thank you for these detailed suggestions, we will make these corrections!
>
> **m3) L217, "8 x 14 x 14" -> "8 x 16 x 16", since you extract 16 x 16 patches from each frame?**
>  We use non-overlapping spatial patches of size 16x16. Since the input frames are of dimension 224x224, it means that the number of spatial patches is 224/16 x 224/16 = 14x14.
>
> **L1) I suggest updating the title, ".. for Multimodal Fusion" since this work limits to bi-modal and audio-visual classification tasks as the authors mentioned in the Conclusion.**
> Our method is not limited to only two modalities, as demonstrated by our additional experiments with RGB, optical flow and spectrograms (please refer to Q3 in the general response for the experiment and discussion). Furthermore, note that our use of fusion tokens allows us to efficiently scale to a larger number of modalities.
>
> **L2) Since the authors evaluate the method using macro metrics for the task, I suggest toning down to "most relevant" -> "relevant" and "only share" -> "share". The current explanation seems to be too strong without sufficient evidence.**
> We agree with this point and will update the paper with your suggestion. In addition, we also extracted some visualisation heatmaps rolled back right down to the inputs which analyze the effect of the bottleneck tokens in focusing on relevant information. See Q2 in the general response where we analyze these visualisations.
>
> **L3) Although the attention bottlenecks have the advantage of computational cost, this is sensitive to the fusion layer. This means if we adjust for the smaller or larger dataset, the hyperparameter would be critical to its performance.**
> Thank you, we will update the "Limitations" section as suggested.

---

### Official Review · Reviewer_yUDq · 2021-07-16

**Rating:** 7
**Confidence:** 4

**Summary:**

The authors propose a new transformer architecture MBT. The MBT uses additional learnable tokens to summarize multimodal information between different modalities and an attention modification to process the constructed multimodal sequences efficiently. They perform ablations on fusion strategies and comparison with related models on audio-visual datasets (such as AudioSet, Epic Kitchen 100, VGGSound). They show improved results on most benchmarks.

**Limitations And Societal Impact:**

The authors have adequately addressed the limitations and potential negative social impact of their work.

**Main Review:**

- The Proposed MBT model shows improved results on most of the benchmarks. However, there is a missing baseline. ViViT [1] can perform better on Epic Kitchens 100 as a video only model. Thus additional experiments are required to show that it improves state-of-the-art and it is worth an additional compute.

  - ViViT-L/16x2 Fact. Encoder outperform MBT using only Video on Verb (66.4 > 64.8 (A,V) > 62. (V)), Action (44 > 43.4 (A, V) > 40.7 (V)). The ViViT is only worse versus MBT (A,V) on Noun (56.8 < 58.0) but MBT's visual (V) version is a bit worse (56.8 > 56.4).
  - ViViT-B/16x2 Fact Encoder does on action better (43.7 > 43.4 (A, V) > 40.7 (V))
  - [1] Arnab, Anurag, et al. "Vivit: A video vision transformer." *arXiv preprint arXiv:2103.15691* (2021).

- There are only two modal video-audio experiments. It is not clear how this idea is versatile to different multimodal scenarios (e.g., any combination of Vision, Audio, Text, Flow; coarse and fine information).

### Minor

- There is no attempt to understand the bottleneck token in terms of multimodal relations.

- Multimodal cross attention (MCA) in section 3.2.2 is related to Co-Attention previously shown by the following work [2].

  - [2] Hendricks, Lisa Anne, et al. "Decoupling the role of data, attention, and losses in multimodal transformers." *arXiv preprint arXiv:2102.00529* (2021).

- Is the order to update the bottleneck tokens essential for performance in section 3.2.3 (e.g., with visual or audio first)? Have you explored other ways how to utilize bottlenecks tokens? Currently, the updates are factorized and asymmetric.

- The tables with comparison do not show backbone architecture, or the number of parameters, or FLOPs. It is hard to understand trade-offs.

- There is no backbone scaling study or transfer learning experiments.

### Originality:

- The idea of multimodal fusion via attention bottlenecks tokens is somewhat novel. It is an adaptation of a classification token to a multimodal application. The modified multimodal attention module is novel.

### Quality:

- Overall it is good work. Specifically, the ablation studies are mostly extensive.

### Clarity:

- The manuscript is well written. Tables with comparison can be improved (see Minor).

### Significance:

- The results are promising. However, a unimodal video model such as ViViT can achieve better results on Epic Kitchens 100. Thus it is not clear that it is state-of-the-art and additional experiments are needed.

**Time Spent Reviewing:**

9

---

> ### Author Response · Authors · 2021-08-10
> **Response to Reviewer yUDq**
>
> Thank you for your constructive comments and positive review.
>
> **There are only two modal video-audio experiments. It is not clear how this idea is versatile to different multimodal scenarios (e.g., any combination of Vision, Audio, Text, Flow; coarse and fine information).**
> There is nothing in our method that is modality specific, or in our formulation that restricts the model to only two modalities. We provide an experiment with RGB, flow and spectrograms to demonstrate this, please see Q3 in the general response for the results and a discussion.
>
> **There is no attempt to understand the bottleneck token in terms of multimodal relations.**
> Thanks for this suggestion. We visualized attention heatmaps rolled back down to the inputs, see Q2 in the general response, where we analyze these visualizations.
>
> **Multimodal cross attention (MCA) in section 3.2.2 is related to Co-Attention previously shown by the following work [2].**
> Yes we agree, and will cite this paper in the related works. Thank you for pointing this out.
>
> **Is the order to update the bottleneck tokens essential for performance in section 3.2.3 (e.g., with visual or audio first)? Have you explored other ways how to utilize bottlenecks tokens? Currently, the updates are factorized and asymmetric.**
> This is a great suggestion. We ran this experiment on mini-Audioset, please see the results provided in Q1 in the general response. We find that the order is not essential at all, and in fact the performance is very similar to a symmetric update.
>
> **The tables with comparison do not show backbone architecture, or the number of parameters, or FLOPs. It is hard to understand trade-offs.**
> Thank you for this suggestion, We have attempted to add these details for Audioset (Table 1 in the main paper) below, however we note that previous works do not always report the number of parameters and FLOPs. While Perceiver [29] reports 707.2 GFLOPs and 44.9M parameters for their best model on Imagenet (Table 6, Appendix B), they do not provide these for AudioSet.
>
> | Model                  | Training Set | **Backbone** | **GFLOPs** | **Params** | A only | V only | AV Fusion |
> |------------------------|--------------|----------|-------|--------|--------|--------|-----------|
> | GBlend [58]            | MiniAS       |  ResNet3D-50       |   110.1 x 30    |   -     | 29.1   | 22.1   | 37.8      |
> | GBlend [58]            | FullAS-2M    |   ResNet3D-50     |    110.1 x 30   |    -    | 32.4   | 18.8   | 41.8      |
> | Attn Audio-Visual [19] | FullAS-2M    |   ResNet-152     |    -    |    -     | 38.4   | 25.7   | 46.2      |
> | Perceiver [29]         | FullAS-2M    |    GPT-2      |   -     |    -     | 38.4   | 25.8   | 44.2      |
> |------------------------|--------------|----------|-------|--------|--------|--------|-----------|
> | MBT                    | MiniAS       |    ViT-B      |    117.31 x 8   |   173.2M     | 31.3   | 27.7   | 43.9      |
> | MBT                    | AS-500K      |   ViT-B       |   117.31 x 8    |   173.2M     | 44.3   | 32.3   | 52.1      |
>
>
> **There is no backbone scaling study.**
> We performed a study using three standard ViT backbones, ViT-Small, ViT-Base and ViT-Large, for 2 of our datasets, Audioset-mini and VGGSound. We report results for RGB+Spec fusion with our best MBT model. The best performance is achieved with ViT-B, and we hypothesise that this is because the large number of parameters in ViT-L leads to overfitting. We will add this table to the paper.
>
> |  | **Audioset-mini** | **VGGSound** |
> |-------|-------|-------|
> | ViT-S | 38.2 | 59.0 |
> | ViT-B |  43.3 | 64.1 |
> | ViT-L | 42.2 | 61.4 |
>
> **There are no transfer learning experiments.**
> We used checkpoints pretrained on VGGSound, Kinetics400 (K400) and AS-500K and finetuned on Audioset-mini and VGGSound (note we use a ViT-B backbone for these experiments, and report results for RGB+Spec fusion with our best MBT model.). While Kinetics400 pretraining gives a slight 0.7% mAP boost on AS-mini, VGGSound initialisation gives a substantial 3% mAP boost over Imagenet Initialisation. On VGGSound, AS500K pretraining gives a more modest boost of 1.2% Top 1 Acc, while Kinetics pretraining does not help (expected as VGGSound is a larger dataset). Results are provided in the table below.
>
> |  | **Audioset-mini** | **VGGSound** |
> |-------|-------|-------|
> | ImageNet init. | 43.3 | 64.1 |
> | VGGSound init. | 46.6 | N/A |
> | K400 init. | 44.0 | 64.0 |
> | AS-500K init. | N/A | 65.3 |
>
> **ViViT [1] can perform better on Epic Kitchens 100 as a video only model.**
> There are a number of differences to our reported results and those used by ViViT.  ViViT uses a factorized spatial-temporal encoder, while we use a standard unfactorised backbone. The ViViT numbers are also reported with a larger model (ViT-L) than ours (ViT-B), and this large model is pretrained on Kinetics, which we do not use. We would also like to highlight that the goal of our paper is multimodal fusion, and our contributions are complementary to the choice of visual backbone – indeed we could replace our backbone with more recent transformer backbones and we would expect performance to increase.
>
> We also note that ViViT is currently unpublished work that was released on arXiv two months before the NeurIPS deadline. According to NeurIPS policy, lack of comparison to such works ''will not result in rejection''.

---

> > ### Comment · Reviewer_yUDq · 2021-08-15
> > **Rebuttal Response**
> >
> > Thank you for answering most of my questions and putting a good amount of effort into improving your manuscript. I don't have concerns. I think it is a good paper, and I recommend acceptance (7).
> >
> > Further ablation with different factorization as in ViViT can make your work better.

---

### Official Review · Reviewer_x2ZW · 2021-07-16

**Rating:** 7
**Confidence:** 5

**Summary:**

The authors propose a new Multimodal Bottleneck Transformer (MBT) for audio-visual learning. Taking image and spectrogram patches as inputs, MBT uses self and cross-attention operations to model both unimodal and cross-modal information at multiple layers in the network. Notably, it adopts a small set of ‘bottleneck’ latent units to force the model to collate and condense the most relevant information in each modality and only share what is necessary. Extensive ablation studies and experiments are conducted on multiple audio-visual classification benchmarks including Audioset, Epic-Kitchens, and VGGSound.

***Post-Rebuttal***

I've read comments from fellow reviewers and the rebuttal from the authors.

Thank the authors for considering my comments and responding to my questions. The rebuttal successfully addressed my major concerns. Thus, I would like to keep my initial positive rating.

The authors should add the missing details, discussion, and new results in the rebuttal to the paper.

**Limitations And Societal Impact:**

Yes, the authors discussed the limitations and Broader Impact.

**Main Review:**

Pros:

+ The proposed model is new and interesting. Two distinct modalities are integrated by a unified multimodal transformer architecture: MBT.  It uses self and cross-attention mechanisms to model both unimodal and cross-modal information at multiple layers. With the bottleneck units, the network becomes more effective and efficient.

+ Thorough ablation studies and experiments on multiple audio-visual classification benchmarks including Audioset, Epic-Kitchens, and VGGSound are provided.  Ablation studies can validate the effectiveness of the fusion design and the proposed method obtains strong results on the benchmarks.

Overall, I like the proposed method and appreciate the thorough experiments. But, I still have some questions and hope the authors can clarify them.

Cons:

- A single visual scene is usually grouped together and has strong local patterns, and multiple visual scenes are usually non-overlapping. I think that is one of the main reasons why convolution, patch-based nonlocal/transformer networks work. However, different from images, a single sound source contains diverse isolated time-frequency patterns; different audio sources can occupy the same frequency at the same time; a single audio source can cover almost all frequency ranges at a time (dog barking sound).  Thus, I worry that these isolated patches might not be able to fully reflect the correct semantic of sound sources. Could the authors provide more discussion and analysis on the validity of audio transformers? Are there any failure examples due to using spectrogram patches?

- To reduce the computation complexity of pairwise attention, fusion bottleneck tokens are introduced. However, it is not clear to me how to initialize these fusion bottleneck tokens. Only the updating method is provided in the paper.

- The authors note that bottlenecks can force the model to collate and condense the most relevant information in each modality and only share what is necessary. However, without any visualizations, the statement is not fully convincing. The authors should provide some examples (e.g., visualizing attention weights) to visually demonstrate the capacity of the bottleneck attention.






**Time Spent Reviewing:**

3

---

> ### Author Response · Authors · 2021-08-10
> **Response to Reviewer x2ZW**
>
> Thank you for your constructive comments and positive review.
>
> **Could the authors provide more discussion and analysis on the validity of audio transformers?**
>
> This is an interesting point. We note that CNNs (with their strong locality inductive biases) have been applied to spectrograms with great successes in the past (e.g. [1*, 2*, 3*]). While it is true that our transformer architecture (similar to AST [20]) is doing a local tiling up front, the transformer architecture can explicitly model interactions between all pairs of patches (including non-adjacent ones) from the very first layer. Unlike CNNs, the "receptive field" of a transformer layer therefore encompasses the entire input at each layer. To investigate the patch choice, we also ran experiments with 1D tiling (i.e. temporal patches with the full frequency band in each tile, in a manner similar to wav2vec 2.0 [4*]) and found this performs ~4% mAP worse than 2D patches in apples-to-apples comparison using large scale audio pretraining. This is consistent with past CNN-based sound classification results that show 2D architectures are generally superior (a broad comparison can be found in Tables 11 and 12 in [2*]). It is also interesting that both CNNs and transformers models have no problem recognizing mixtures of overlapping sound events (i.e. there is no gain when we first apply sound separation and classify each source separately). There has been an interesting line of work on alternative convolutional architectures that attempt to take advantage of other signals like harmonic structure in sound (see [5*]), as well as those directly modeling the raw waveform (e.g. [6*, 7*]), however some time windowing is always needed for computational tractability. We will add this discussion to Sec. 2 of the paper.
>
> **It is not clear to me how to initialize these fusion bottleneck tokens.**
>
> Thanks for spotting this! For the fusion bottleneck tokens we use Gaussian initialization with mean of 0 and standard deviation of 0.02, in a manner similar to the way positional embeddings are initialized following BERT [11] and ViT [12] public code. We will add this to the paper.
>
> **The authors should provide some examples (e.g., visualizing attention weights) to visually demonstrate the capacity of the bottleneck attention.**
>
> This is a great idea. We visualised attention heatmaps rolled back right down to the inputs, discussed in Q2 in the general response, and find that the bottlenecks in MBT force the attention to be localised to smaller regions of the images compared to vanilla attention.
>
>
> [1*] Hershey, Shawn, et al. "CNN architectures for large-scale audio classification." 2017 ieee international conference on acoustics, speech and signal processing (icassp). IEEE, 2017.
>
> [2*] Kong, Qiuqiang, et al. "Panns: Large-scale pretrained audio neural networks for audio pattern recognition." IEEE/ACM Transactions on Audio, Speech, and Language Processing 28 (2020): 2880-2894.
>
> [3*] Gong, Yuan, Yu-An Chung, and James Glass. "PSLA: Improving Audio Tagging with Pretraining, Sampling, Labeling, and Aggregation." arXiv preprint arXiv:2102.01243 (2021).
>
> [4*] Baevski, Alexei, et al. "wav2vec 2.0: A Framework for Self-Supervised Learning of Speech Representations." Advances in Neural Information Processing Systems 33 (2020).
>
> [5*] Zhang, Zhoutong, et al. "Deep audio priors emerge from harmonic convolutional networks." International Conference on Learning Representations. 2019.
>
> [6*] Sainath, Tara N., et al. "Learning the speech front-end with raw waveform CLDNNs." Sixteenth Annual Conference of the International Speech Communication Association. 2015.
>
> [7*] Zeghidour, Neil, et al. "Leaf: A learnable frontend for audio classification." arXiv preprint arXiv:2101.08596 (2021).

---

### Official Review · Reviewer_SoXm · 2021-07-21

**Rating:** 7
**Confidence:** 4

**Summary:**

This paper conducts an experimental study of different fusion methods for multimodal audiovisual transformers, and proposes a new fusion mechanism. They mix ordinary transformer layers with new fusion attention layers. These new layers perform cross-attention, in which one modality is used and the other is used as the key/values. This takes places using special fusion tokens, which have a function similar to the classification tokens in ordinary transformers. They find that (1) mid-fusion can obtain better results than early/late fusion in both their model and "vanilla" fusion, (2) attention bottlenecks improve performance, (3) they outperform the very recent, state-of-the-art Perceiver model on AudioSet classification. They also conduct additional experiments on VGG-Sound and Epic Kitchens, in which they compare with other state-of-art models. The supplement contains several other datasets.

**Limitations And Societal Impact:**

These were addressed adequately in the conclusion.

**Main Review:**

Positives:
- The attention bottleneck mechanism is simple and has a compelling motivation over "vanilla" fusion.
- The idea of examining the effectiveness of early/late fusion has not, as far as I know, been examined in the context of transformer models.
- This paper looks at fusion designs in a more systematic way than most prior work, which is enabled by the simple design of their transformer architecture.

Negatives:
- The cross-attention function MCA() is seemingly asymmetric (L122), using only X as the query in the multi-headed attention. Are both directions of attention used in Eq. 8 and Eq. 9, thereby allowing by the bottleneck and the modalities to be updated?
- The comparison to Perceiver is not quite apples-to-apples, due to the use of more extensive augmentation (as well as the different train set). Though it might not be essential, it would clarify the comparison if a version of Perceiver were trained using the same setup.
- There is no direct comparison to AVSlowFast [49], despite training on similar datasets (such as Kinetics-400 in the supplement), and having similar goals.

Overall:
This paper presents a very interesting analysis of different fusion strategies in transformers. The bottleneck idea is simple and compelling. The model obtains large gains from fusion on a variety of datasets. I therefore lean towards acceptance.

After the discussion:
- The rebuttal clarified my question about the asymmetric updates.
- The response about comparisons seems reasonable.
Overall, I still lean towards acceptance.

**Time Spent Reviewing:**

3.5

---

> ### Author Response · Authors · 2021-08-10
> **Response to Reviewer SoXm**
>
> Thank you for your constructive comments and positive review.
>
> **Are both directions of attention used in Eq. 8 and Eq. 9, thereby allowing both the bottleneck and the modalities to be updated?**
>
> Yes, both directions are used, which means that the bottlenecks and the single modality tokens are updated. We will clarify this in the paper. While the bottleneck attention is applied both ways, we initially applied it with an asymmetric ordering, wherein one modality is updated before another. See general response Q1 for an experiment with symmetric updates as well as changing the ordering of the asymmetric update.
>
> **Comparison to Perceiver.**
>
> We note that the Perceiver model/code has not been released online, but once it is we will train the model with our exact experimental setup for comparison. We tried to mimic the augmentation settings of Perceiver as closely as possible, and found that without the additional augmentation and regularisation, our model achieves an mAP of 50.4 trained on the AS-500K split (which is a strict subset of the training set for Perceiver). This comfortably outperforms Perceiver (note the arXiv paper has been updated with a lower number [44.2 mAP] due to a bug in their evaluation code). We will add this ablation to the paper.
>
> **There is no direct comparison to AVSlowFast [49].**
>
> Thanks for noticing this, our model outperforms the best AVSlowFast-101 model on Kinetics by 2% in top-1 accuracy (80.8% vs 78.8%), we will add this to the Kinetics table in the appendix.

---

### Author Response · Authors · 2021-08-10
**General Response common to all Reviewers**

We thank all reviewers for their positive comments and feedback. We first address some general points here and refer to additional visualisations and figures at the anonymous link here: https://11137mbt.github.io/11137/.

**Q1. Asymmetric Bottleneck Updates:**

Yes, our bottleneck updates were performed asymmetrically. We experimented with changing the order of the asymmetric updates, i.e. updating the spectrogram latent units before the RGB latent units. We also performed an experiment with a symmetric bottleneck update, implemented as:

$ [z_{i}^{l+1} || z_{fsn_i}^{l+1}] = \text{Transformer}( [ z_{i}^{l} || z_{fsn}^{l} ]; \theta_{i}  ) $

$ z_{fsn}^{l+1} = \\text{Avg}\_{i}( z_{fsn_i}^{l+1} ) $

Where $i$ is a parameter that specifies which modality is being used.
The results on Audioset-mini are below (note we perform 3 runs and average the mAP):

| | RGB first | Spec first | Symmetric update |
|--------|-------|-------|-------|
| Audioset-mini mAP |  43.42$\pm$0.19 | 43.23$\pm$0.12 | 43.66$\pm$0.26 |

The results indicate that performance is robust to the different variations. We will update the paper to include the symmetric update as default, and add this ablation study to the appendix.

**Q2. Attention Visualization.**

This is a great idea. We visualised attention heatmaps rolled from the CLS tokens right down to the inputs (Fig. 1 of this anonymous link: https://11137mbt.github.io/11137/), and compared a vanilla attention model with MBT. Attention heatmaps are obtained using Attention Rollout [1*], and for RGB frames we show the attention maps summed over all the frames in the video clip. On the RGB images, it is interesting to note that for both models the attention is particularly focused on moving sound source regions in the video, eg. the fingertips on the piano, the hands on the string instrument, faces of humans. We find that the bottlenecks in MBT further force the attention to be localised to smaller regions of the images (i.e the mouth of the baby on the top left and the mouth of the woman singing on the bottom right). For cases where the sounding object is already localised to a narrow region of the image clearly (bottom row), the differences between vanilla fusion and MBT are less apparent. We will add these visualisations as a figure to the paper. For completeness, we also show visualisations on the spectrograms, however we note that these are less interpretable.

**Q3. Extension to more than 2 modalities.**

This is a good suggestion. As shown in the equations in Q1, our formulation is generic to the type and the number of modalities. To demonstrate this experimentally, we extract optical flow using the TVL1 algorithm [2*] and train MBT on flow only, RGB+flow (which historically has been a common modality combination for video recognition), as well as RGB+flow+spectrogram. (See, Fig. 2, at this anonymous link: https://11137mbt.github.io/11137/ for a diagram of MBT with 3 modalities). The results are provided below. We note that for a sound classification dataset like AudioSet, RGB + Flow provides a 1% mAP boost over RGB only, but once we add spectrograms, the gains are reduced significantly. Given many of the sound categories are not motion-related, this is expected for this dataset. Because optical flow is derived from RGB, it should also intuitively provide less of a complementary signal compared to audio. We will include this study for more datasets in the paper, particularly for action focused video datasets like Epic-Kitchens where flow might be a more valuable input.

| | RGB only | Flow only | RGB + Flow | RGB + Spectrogram | RGB + Flow + Spectrogram |
|--------|-------|-------|-------|-------|-------|
| Audioset-mini mAP |  27.7 | 8.67 | 28.8 | 43.9 | 44.0 |

This demonstrates the versatility of MBT, and that for other tasks/datasets where more modalities are available, our formulation can naturally scale.

[1*] Samira Abnar and Willem Zuidema. Quantifying attention flow in transformers. arXiv preprint
arXiv:2005.00928, 2020.

[2*] Pérez, Javier Sánchez, Enric Meinhardt-Llopis, and Gabriele Facciolo. "TV-L1 optical flow estimation." Image Processing On Line 2013 (2013): 137-150.

---

### Decision · Program_Chairs · 2021-09-27

**Decision:**

Accept (Poster)

**Comment:**

The reviewers unanimously recommend an acceptance. They acknowledged that the proposed attention bottleneck module is simple and demonstrated to be effective by extensive experiments and ablation analyses. They also appreciated an empirical exploration of different fusion mechanisms in Transformer models. The initial reviews raised some concerns about insufficient experiments and asked clarifying questions, and the rebuttal successfully cleared up the questions. Overall, this is a nice paper addressing an important problem of modeling multimodal data and proposes a simple method based on Transformer architectures, which is a timely topic for this conference.